# An Unusual Case of *Listeria monocytogenes*-Associated Rhombencephalitis Complicated by Brain Abscesses in Italy, 2024

**DOI:** 10.3390/idr18010005

**Published:** 2026-01-04

**Authors:** Maria Gori, Giorgia Orsani, Carlotta Ortelli, Erika Scaltriti, Luca Bolzoni, Luigi Vezzosi, Silvia Bianchi, Clara Fappani, Daniela Colzani, Antonella Amendola, Danilo Cereda, Laura Marzorati, Stefano Pongolini, Elisabetta Tanzi

**Affiliations:** 1Department of Health Sciences, Università degli Studi di Milano, 20133 Milan, Italy; 2Department of Neurology, University of Milano-Bicocca, San Gerardo Hospital, ASST Monza, 20900 Monza, Italy; 3Risk Analysis and Genomic Epidemiology Unit, Istituto Zooprofilattico Sperimentale della Lombardia e dell’Emilia-Romagna, 43126 Parma, Italy; 4Directorate General for Health of the Lombardy Region, 20124 Milan, Italy

**Keywords:** listeriosis, cerebral abscesses, surveillance, foodborne pathogens

## Abstract

Background/Objectives: *Listeria monocytogenes* (*Lm*) is an extremely rare cause of brain abscesses, accounting for 1–10% of neurolisteriosis cases reported in the literature, associated with high mortality (approximately 23%). Data on diagnosis, management, and treatment is scarce. We report a case of listerial brain abscesses in an elderly patient in Italy who experienced progressively worsening bilateral ptosis. Methods: Diagnostic evaluation included neuroimaging, blood cultures, and microbiological investigations, followed by antimicrobial treatment according to available evidence. The isolated Lm strain underwent whole genome sequencing. Dietary history was also collected. Results: Positive early blood cultures were pivotal in identifying Lm as the aetiological agent. Neuroimaging revealed brain abscesses consistent with neurolisteriosis. The clinical course was complicated by pneumonia and opportunistic co-infecting pathogens, and despite adequate treatment according to the available literature, the outcome was fatal. Genomic characterisation revealed that the patient was infected with an strain belonged to the sequence type 206 and clonal complex 14, described as hypervirulent. The patient reported consuming several foods known to be associated with an increased risk of listeriosis. Conclusions: This case highlights the challenges involved in diagnosing and managing listerial brain abscesses, particularly in elderly patients. Even when the primary central nervous system infection is under control, the prognosis may be significantly impacted by comorbid conditions and hospital-related complications rather than the infection itself. Our findings underscore the need for improved preventive strategies and targeted risk communication regarding high-risk foods, particularly among elderly populations.

## 1. Introduction

*Listeria monocytogenes* (*Lm*) is a facultative intracellular pathogen that is widespread in the environment and is responsible for human listeriosis, a rare but life-threatening foodborne disease that most often presents as sepsis or central nervous system (CNS) manifestation, especially in neonates, old, pregnant, and immunocompromised individuals [1].

As an intracellular pathogen, *Lm* has an arsenal of virulence factors that enable it to invade host cells and evade the host immune response. Neuro-invasion and translocation across the blood–brain barrier are primarily mediated by surface proteins called internalins (InlA, InlB, and InlF) [1]. This process typically occurs in the context of a systemic disease through direct invasion of endothelial cells from bacterial dissemination via the bloodstream or the ‘Trojan horse mechanism’, whereby bacteria are transported as cargo in infected circulating leukocytes [1,2].

Meningitis and meningoencephalitis are the most common CNS manifestations, whereas cerebral abscesses occur extremely rarely, accounting for 1–10% of CNS *Lm* infections and 1% of all *Lm* infections [3,4,5]. To date, fewer than one hundred cases of brain abscesses related to *Lm* infection have been reported in the literature [3,4,5,6,7,8,9,10], with only three cases occurring in Italy [3,8]. Here, we present an additional case involving an elderly patient.

## 2. Case Description

On 31 January 2024, a 93-year-old patient attended an Emergency Department in Lombardy, Northern Italy, with fall-related head trauma due to the inability to open the eyelids. Two days before admission, the patient experienced blurred vision and progressively worsening bilateral ptosis, up to the complete closure of the eyelids.

Medical history was significant for hypertension. Upon arrival at the hospital, the patient weighed 65 kg. Recent upper respiratory tract, dental, or ear infections were denied.

Haematological tests at admission were remarkable for elevated levels of C-reactive protein (4.28 mg/dL; reference range < 0.50 mg/dL), creatine kinase (276 U/L; reference range < 170 U/L), and monocytes (14.3%; reference range 4.0–11.0%).

Ophthalmic evaluation diagnosed a deficit of elevation in adduction in the right eye and a deficit of elevation and nystagmus in the left eye. The patient was transferred to the Neurology Department based on the suspicion of myasthenia gravis. Electromyography and anti-acetylcholine receptor antibodies were performed.

Electromyography results were normal, and blood results were negative for anti-acetylcholine receptor antibodies.

On 4 February 2024, the patient experienced fever up to 38.5 °C. Empirical antibiotic therapy with intravenous ceftriaxone (2 g) twice daily was administered, and peripheral venous blood cultures were taken. Chest X-ray revealed bibasal pleural effusion. Magnetic resonance imaging (MRI) of the brain with contrast medium showed the presence of multiple ring-enhancing lesions located in the brainstem, compatible with brain abscesses (Figure 1 and Figure 2).

Based on the radiological findings and the need to ensure coverage for a broad spectrum of potential pathogens (e.g., *Tropheryma whipplei*), starting from 6 February 2024, the therapy was shifted to broad-spectrum therapy with intravenous meropenem (2 g three times daily) and trimethoprim-sulfamethoxazole (160–800 mg three times daily), using commercially available formulations.

Given the high-risk location of the abscesses and the advanced age and frailty of the patient, lumbar puncture and biopsy were not carried out. This decision followed multidisciplinary evaluation involving neurology, infectious diseases, and radiology specialists, who judged the procedures to be unsafe and unlikely to change management.

Blood cultures were incubated using the BactAlert system (bioMérieux, Marcy-l’Étoile, France) and yielded positive results three days after collection. *Lm* was identified using MALDI-TOF mass spectrometry (MALDI Biotyper microflex LT, Bruker Daltonics, Bremen, Germany): neurological diagnosis was rhombencephalitis due to *Lm*.

Antibiotic susceptibility testing was performed using the VITEK 2 Compact system (bioMérieux, Marcy-l’Étoile, France). Interpretations of minimum inhibitory concentrations (MICs) were made according to the European Committee on Antimicrobial Susceptibility Testing (EUCAST) guidelines [11]. The isolate was susceptible to ampicillin, meropenem, penicillin G, erythromycin, and trimethoprim-sulfamethoxazole, according to the EUCAST breakpoints.

Brain MRI was repeated two weeks after the start of the antimicrobial therapy, showing a reduction in number and size of cerebral abscesses. The patient experienced slight clinical improvement, with a reduction in left eye ptosis. Meropenem was discontinued on 20 February 2024, after two weeks of treatment, to simplify the therapy, and plans were made to transfer the patient to a nursing home.

Trimethoprim-sulfamethoxazole was administered for a total of four weeks.

The patient reported having eaten several possible high-risk foods for *Lm* contamination in the two months preceding the onset of symptoms, i.e., smoked salmon, fresh salami, gorgonzola cheese, and sliced veal with a mayonnaise-like tuna sauce. Food samples were no longer available for microbiological sampling.

Starting from 29 February 2024, the patient experienced a sudden worsening of symptoms, with desaturation, fever of 38.8 °C, shaking chills, and progressive deterioration of consciousness. Chest X-rays documented right pneumonia, and haematological examinations revealed a slight increase in procalcitonin (0.65 ng/mL; reference range 0.00–0.50 ng/mL) and neutrophil (85.3%; reference range 40.0–75.0%) levels. Based on the suspicion of septic shock, blood and urine cultures were taken. Supportive treatment with oxygen with a venti-mask was performed. Crystalloid infusion, paracetamol, and non-steroidal anti-inflammatory drugs were also administered. Intravenous antimicrobial therapy with commercially available formulations of meropenem (1 g three times daily) and linezolid (600 mg twice daily) was initiated. Antimicrobial therapy was suspended the following day after the decision to start palliative care.

The patient died on 2 March 2024.

Blood and urine culture results were obtained after death and revealed the presence of *Candida albicans* and *Enterococcus faecalis*, respectively. *Lm* was no longer present.

The *Lm* isolate retrieved from the patient’s blood cultures was sent to the regional reference laboratory for whole-genome sequencing (WGS) [12]. Genomic DNA was extracted using a Maxwell^®^ HT 96 gDNA Blood Isolation System (Promega, Madison, WI, USA). WGS was conducted on a Nextseq 550 system (Illumina, San Diego, CA, USA). Sequence Type (ST), Clonal Complex (CC), and PCR serogroup were deducted in silico using Institut Pasteur’s BIGSdb-*Lm* [13]. The isolate belonged to ST206/CC14 and PCR serogroup 1/2a. We performed comparative genomics to analyse allele distances in core-genome multi-locus sequence typing scheme from BIGSdb-*Lm* [13] and SNPs using the CFSAN SNP Pipeline v.2.1.1 (Center for Food Safety and Applied Nutrition, US Food and Drug Administration, College Park, MD, USA) [14]. The strain was not related to any other strain in Lombardy from 2021 to date.

This case report was prepared in accordance with the CARE guidelines, and the completed CARE checklist is provided as Appendix A.

## 3. Discussion

Brain abscesses are a rare manifestation of neurolisteriosis, with fewer than one hundred cases reported in the literature to date [3,4,5,6,7,8,9,10]. Due to their rarity, there are currently no standardised diagnostic or therapeutic guidelines for managing this condition [3,9].

In a large French cohort of over 800 listeriosis cases, Charlier et al. found that brain abscesses were present in only 2% of neurolisteriosis cases—defined as clinical neurological symptoms with *Lm* isolated from the cerebrospinal fluid (CSF), brain abscesses, or blood cultures [15]. Although lumbar puncture and abscess biopsy could not be performed on our patient due to their advanced age and the associated anatomical risks, the diagnosis of neurolisteriosis was supported by positive blood cultures and neuroimaging findings that were consistent with the criteria outlined by Charlier et al. [15].

Positive blood cultures are a hallmark of listerial brain abscesses. Carneiro et al. reported blood culture positivity in 79% of published cases compared to 55% for CSF or abscess specimens [5].

Similarly, Eckburg et al. found that 86% of patients had positive blood cultures, whereas only 38% had positive CSF cultures [4]. This contrasts sharply with brain abscesses of other bacterial aetiologies, where bloodstream invasion is rare and positive blood cultures are found in only 11% of cases. The relatively high frequency of *Lm* bacteraemia is consistent with its pathogenesis, involving haematogenous dissemination following the ingestion of contaminated food [1,2,4].

Unlike abscesses caused by other pathogens, listerial abscesses are not usually associated with local extension from adjacent foci, such as otitis or dental infections. To date, no such cases have been reported [4].

The predilection of *Lm* for posterior fossa structures is well documented, particularly with regard to rhombencephalitis, as was observed in our patient. However, supratentorial and multicentric lesions have also been described [3,4,5,15]. Brainstem localisation likely reflects specific bacterial tropism and neuro-invasion mechanisms, including translocation across the blood–brain barrier and neuronal migration [2].

Rhombencephalitis represents a heterogeneous clinical syndrome with a broad differential diagnosis, including both infectious and non-infectious aetiologies. Among infectious causes, *Lm* is consistently reported as the most frequent pathogen, particularly in adults and elderly patients [16]. In cohorts of human listeriosis, brainstem involvement in the form of rhombencephalitis has been described in up to approximately 20–25% of neurolisteriosis cases, highlighting the marked neurotropism of this pathogen [17]. Other infectious aetiologies of rhombencephalitis are reported far less frequently and are mostly described in isolated case reports or small case series. These include viral infections such as herpes viruses—particularly human herpes virus 6 and varicella-zoster virus—even in immunocompetent hosts [18,19] as well as emerging arboviruses such as Powassan virus [20]. Bacterial causes other than *Lm* are uncommon but include neuroborreliosis [21] and, more rarely, central nervous system infection due to *Tropheryma whipplei* [22].

Non-infectious aetiologies of rhombencephalitis constitute a clinically relevant but variably reported proportion of cases and include autoimmune and inflammatory disorders, such as Neuro-Behçet’s disease [23], as well as paraneoplastic syndromes associated with neuronal autoantibodies [24]. In the present case, alternative aetiologies were actively considered during the initial diagnostic work-up prior to the identification of the causative pathogen. However, the presence of positive blood cultures for *Lm*, characteristic neuroimaging findings, and the partial radiological response to targeted antimicrobial therapy strongly supported the diagnosis of listerial rhombencephalitis complicated by brain abscesses. Furthermore, the patient’s advanced age, the acute clinical course, and the absence of systemic or neurological features typically associated with autoimmune, inflammatory, or paraneoplastic disorders made non-infectious aetiologies less likely in the clinical context [24].

Our patient’s clinical course, which initially improved under targeted antimicrobial therapy before deteriorating fatally, mirrors the poor prognosis often associated with listerial brain abscesses.

Reported mortality rates range from 27 to 60%, depending on the patient population and treatment approach [4,5,15]. Although a radiological response was observed, the patient ultimately succumbed to complications that were not directly related to the primary *Lm* CNS infection. These complications were likely attributable to age-related frailty and nosocomial superinfections.

The *Lm* isolate from our case belonged to clonal complex 14, which has been described as hypervirulent due to its ability to survive under host immune pressure and its high replicative potential in *Galleria mellonella* infection models [25].

*Lm* should be considered as a triggering agent for brain abscesses in patients with neurological symptoms and blood cultures positive for *Lm*, since early diagnosis and proper treatment could be pivotal to achieving a favourable outcome. However, because of its high mortality rate, this condition often results in poor outcomes. In our case, the patient initially showed partial clinical improvement under targeted antimicrobial therapy. However, the subsequent fatal outcome was likely not directly attributable to the listerial brain abscesses, which had shown radiological improvement. Rather, it appears to have been the result of a combination of advanced age, underlying frailty, and the development of secondary nosocomial infections.

This case illustrates the complexity of diagnosing and managing listerial brain abscesses, especially in elderly patients where, even when the primary CNS infection is partially controlled, the prognosis may be significantly affected by comorbid conditions and hospital-related complications rather than by the infection itself.

## Figures and Tables

**Figure 1 idr-18-00005-f001:**
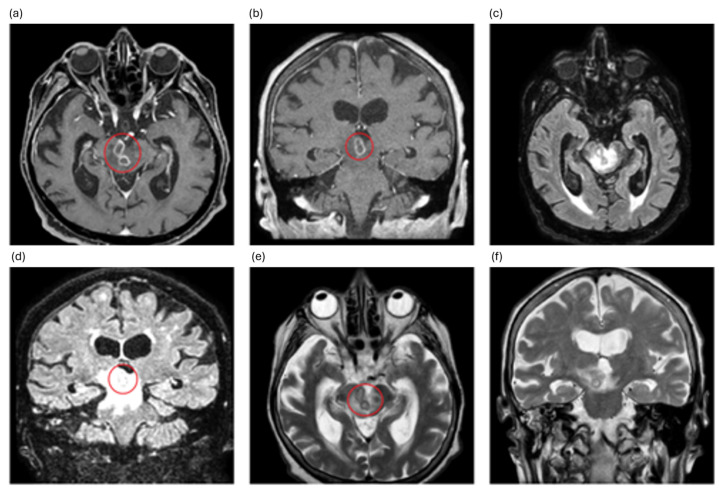
MRI findings at the level of the midbrain, shown across multiple sequences and planes: (**a**,**b**) axial and coronal post-contrast T1-weighted 3D Dixon images show small lesions in the right cerebral peduncle, some of which exhibit peripheral contrast enhancement, consistent with blood–brain barrier disruption; (**c**,**d**) axial and coronal FLAIR images demonstrate heterogeneous hyperintense signal abnormalities in the same region, with a surrounding area of vasogenic oedema extending caudally to the ponto-mesencephalic junction, superior cerebellar peduncles (right > left), and the right middle cerebellar peduncle, and cranially to the subthalamic region (right > left). (**e**,**f**) Axial and coronal T2-weighted images show the extent of the oedema and mass effect, notably on the mesencephalic aqueduct—which appears stenotic—and on the right lateral wall of the third ventricle. Abscess lesions are highlighted with a red circle.

**Figure 2 idr-18-00005-f002:**
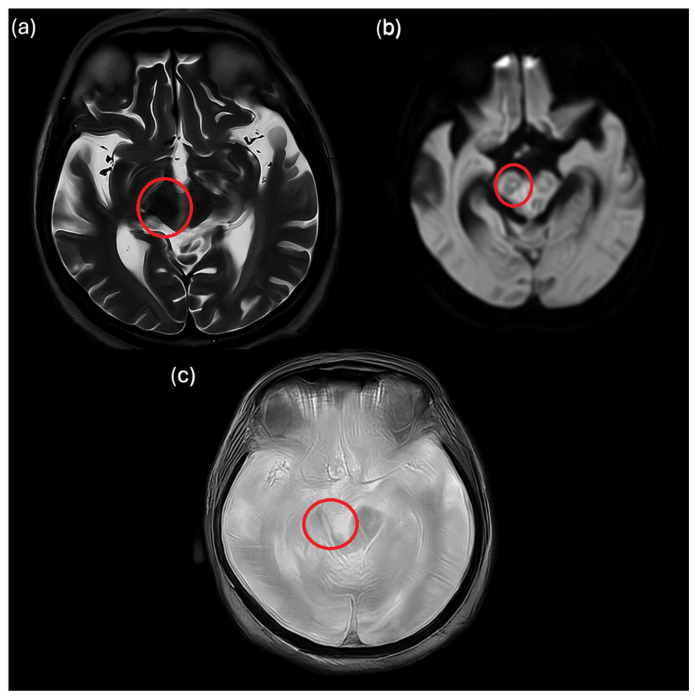
Axial MRI images (**a**–**c**) demonstrate a reduction in both number and size of the previously described mesencephalic lesions, now primarily located in the right cerebral peduncle. These lesions still show restricted diffusion and heterogeneous T2-FLAIR signal. There is a marked decrease in the perilesional vasogenic oedema and in the mass effect on the mesencephalic aqueduct and third ventricle. A slight reduction in overall ventricular size is also observed. Abscess lesions are highlighted with a red circle.

## Data Availability

The data presented in this study are available upon request from the corresponding author.

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
