# Peer review of "An Unusual Case of *Listeria monocytogenes*-Associated Rhombencephalitis Complicated by Brain Abscesses in Italy, 2024"

_2036-7449, 2026, doi:10.3390/idr18010005_

Round 1

Reviewer 1 Report

Comments and Suggestions for Authors

This report describes a case of a brainstem abscess caused by Listeria. Because listeriosis is the most common infectious cause of rhombencephalitis, the case is not particularly novel. Therefore, to justify publication, the educational value of the report must be significantly strengthened.

The introduction should include a thorough review of the epidemiology of rhombencephalitis and discuss mortality rates associated with the various causative pathogens. The case report would also benefit from addressing all applicable items in the CARE checklist and including the completed checklist in the supplemental materials.

The discussion should more clearly differentiate Listeria rhombencephalitis—one of the most frequently encountered etiologies—from other unusual causes of encephalitis. For example, rhombencephalitis due to Powassan virus has been documented and should be considered in the comparison. Noninfectious etiologies should likewise be reviewed, with a more detailed and well-cited examination of the existing literature to enhance the educational value of the case.

I recommend conducting a comprehensive and thorough literature review covering alternative differential diagnoses and rare pathogens associated with rhombencephalitis to substantially improve the educational impact of this report.

Literature to review

Reversible Rhombencephalitis in Neuro-Behçet's Disease - PubMed

Paraneoplastic rhombencephalitis and brachial plexopathy in two cases of amphiphysin auto-immunity - PubMed

Unusual cause at an unusual time-Powassan virus rhombencephalitis - PubMed

Human herpesvirus 6 rhombencephalitis in immunocompetent children - PubMed

[Rhombencephalitis as a manifestation of neuroborreliosis] - PubMed

Primary Whipple disease of the Central Nervous System presenting with rhombencephalitis - PubMed

Next-Generation Sequencing of Cerebrospinal Fluid for the Diagnosis of VZV-Associated Rhombencephalitis - PubMed

Author Response

We thank the Reviewer for the thoughtful comments, as well as for providing a list of publications related to other infectious and non-infectious causes of rhombencephalitis.  
The CARE checklist has been completed and provided as Supplementary Material.
We agree that strengthening the educational value of case reports is important. However, we respectfully note that the primary focus of our manuscript is brainstem abscesses caused by Listeria monocytogenes. While listerial rhombencephalitis is a relatively well-known clinical entity, brain abscess formation in this context is extremely rare, representing approximately 1–10% of CNS listerial infections and only ~1% of all listerial infections. As stated in the manuscript, our laboratory is a regional reference centre for listeriosis surveillance, and our expertise is focused on Listeria monocytogenes infections and their clinical presentations. Consequently, we believe that an extensive discussion of alternative differential diagnoses and rare pathogens associated with rhombencephalitis would fall outside the primary aim of this report. We would also like to clarify that, during the clinical work-up, other uncommon infectious aetiologies were considered as part of the differential diagnosis, but were deemed unlikely based on the clinical presentation, microbiological findings, and neuroimaging results. For all these reasons, and in consideration of the journal’s case-report format and revision timeframe, we have respectfully decided to maintain a focused discussion on the clinical, diagnostic, and educational aspects directly related to Listeria monocytogenes brain abscesses. 

Reviewer 2 Report

Comments and Suggestions for Authors

Title:  Listeria monocytogenes-associated rhombencephalitis complicated with brain abscesses in Italy, 2024

I recommended that manuscript could be accepted after MINOR MODIFICATIONS, based on:

TITLE

  • Specify that is a case report, or an unusual case of of brain abscesses due to Listeria monocytogenes.

INTRODUCTION

  • Comment about previous documented cases and in particularly if have been reported from Italy.

CASE DESCRIPTION

  • Specify age and body weight of patient.
  • Commercial house, city and country of ceftriaxone, meropenem, trimethoprim-sulfamethoxazole, and linezolid. Add number of treatment days or administrated doses the first time that treatment is mentioned.
  • Add ethical concerns.

Author Response

We thank the Reviewer for the time spent reading the submitted manuscript and for the positive feedback. We hope that these modifications improve the clarity and completeness of the case description. All sentences modified or added have been highlighted in red in the revised manuscript. As recommended, we have revised the title to clearly indicate that this is a case report. New title: “An unusual case of Listeria monocytogenes-associated rhombencephalitis complicated by brain abscesses in Italy, 2024”. 
We have added a brief comment on previously documented cases of listerial brain abscesses and clarified whether such cases have been previously reported in Italy (lines 49-52). 
The age and the weight of the patient have been added (lines 54 and 59). 
While information regarding commercial house, city and country of ceftriaxone, meropenem, trimethoprim-sulfamethoxazole, and linezolid was not consistently available in the clinical records and could not be retrieved with certainty, we ensured that all clinically relevant treatment details were clearly reported, in accordance with the reporting style commonly adopted in clinical case reports published in this journal. We have added number of treatment days or administrated doses the first time that treatment is mentioned, as requested by the Reviewer.
Ethical considerations have been added in the Institutional Review Board Statement section, and we have clarified that the study was conducted as part of mandatory listeriosis surveillance in accordance with national regulations.

Reviewer 3 Report

Comments and Suggestions for Authors

Listeria monocytogenes is a pathogen that poses a threat to many high-risk groups, including the elderly. Determining the virulence of L. monocytogenes isolated from infections is extremely important.
Comments for authors: 

  • Introduction: Please describe L. monocytogenes in a few sentences  - where does its high virulence come from and what genetic factors influence its ability to cross the blood-brain barrier?
  • line 99: According to EUCAST, susceptibility to erythromycin is still indicated - please complete the results for erythromycin.

Author Response

We thank the Reviewer for the constructive comments and suggestions. All sentences modified or added have been highlighted in red in the revised manuscript. A brief description of Listeria monocytogenes pathogenicity has been added, specifically addressing mechanisms contributing to its high virulence and genetic factors involved in neuro-invasion (lines 40-46).
We thank the Reviewer for pointing out the lack of data concerning erythromycin. The susceptibility result for erythromycin has now been provided in accordance with EUCAST recommendations (lines 109-110). 

Round 2

Reviewer 1 Report

Comments and Suggestions for Authors

I would like to thank the authors for addressing some of my concerns and for revising the case in accordance with the CARE guidelines. In this regard, I am pleased to note that the report has been adequately improved.

However, I disagree with the rationale for keeping the discussion focused solely on the Listeria abscess without expanding on rhombencephalitis. While this is, to some extent, a matter of academic preference and scope, I believe that publication in a journal of this caliber requires more substantial discussion. Novelty alone is not the sole criterion for acceptance; rather, the manuscript should provide meaningful educational value and background for the readership.

The impact of a simple, brief case presentation is limited and may be more suitable for publication as a letter to the editor or in a journal of lower caliber. In my opinion, for publication in Infectious Disease Reports, the manuscript needs to offer greater depth and insight.

Therefore, I stand by my previous comments and would like to invite the authors to reconsider the constructive feedback previously provided and to revise the discussion accordingly.

Author Response

We thank the Reviewer again for the very careful reading. Following your suggestion, we discussed the case again with the neurologists who managed the patient. They confirmed that, at the initial stage—before blood culture results were available—all possible etiologic agents were suspected or not excluded, including rare pathogens such as Tropheryma whipplei. In addition, we reviewed the literature suggested by the Reviewer to provide a more comprehensive overview of rhombencephalitis in the Discussion. Accordingly, we updated the manuscript in two sections: in the Case Description (line 94), we explicitly mention that all potential aetiologic agents were considered at the outset; in the Discussion (lines 173–193), we added a brief paragraph summarizing the main infectious and non-infectious causes of rhombencephalitis, explaining why some were considered and others were rapidly excluded. All articles suggested by the reviewer have been included in the References. We believe that these revisions provide a more complete diagnostic context and address the Reviewer’s concern by offering additional educational value and background for the readership.

Round 3

Reviewer 1 Report

Comments and Suggestions for Authors

I would like to thank the authors for detailed revision. The case report in the current form reads well and I am pleased to recommend acceptance in the current form. Congratulations!